# Genome-Wide Analysis of the *LATERAL ORGAN BOUNDARIES Domain* (*LBD*) Members in Alfalfa and the Involvement of *MsLBD48* in Nitrogen Assimilation

**DOI:** 10.3390/ijms24054644

**Published:** 2023-02-28

**Authors:** Xu Jiang, Huiting Cui, Zhen Wang, Junmei Kang, Qingchuan Yang, Changhong Guo

**Affiliations:** 1College of Life Science and Technology, Harbin Normal University, Harbin 150025, China; 2College of Grassland Science and Technology, China Agricultural University, Beijing 100193, China; 3Institute of Animal Science, Chinese Academy of Agricultural Sciences, Beijing 100093, China

**Keywords:** *Alfalfa*, *LBD* gene family, evolutionary relationship, expression profiles, nitrogen response

## Abstract

The LATERAL ORGAN BOUNDARIES DOMAIN (LBD) proteins, a transcription factor family specific to the land plants, have been implicated in multiple biological processes including organ development, pathogen response and the uptake of inorganic nitrogen. The study focused on *LBD*s in legume forage *Alfalfa*. The genome-wide analysis revealed that in *Alfalfa* 178 loci across 31 allelic chromosomes encoded 48 unique LBDs (MsLBDs), and the genome of its diploid progenitor *M. sativa* spp. *Caerulea* encoded 46 *LBD*s. Synteny analysis indicated that the expansion of *Alfalfa*
*LBD*s was attributed to the whole genome duplication event. The MsLBDs were divided into two major phylogenetic classes, and the LOB domain of the Class I members was highly conserved relative to that of the Class II. The transcriptomic data demonstrated that 87.5% of *MsLBD*s were expressed in at least one of the six test tissues, and Class II members were preferentially expressed in nodules. Moreover, the expression of Class II *LBD*s in roots was upregulated by the treatment of inorganic nitrogen such as KNO_3_ and NH_4_Cl (0.3 mM). The overexpression of *MsLBD48*, a Class II member, in *Arabidopsis* resulted in growth retardance with significantly declined biomass compared with the non-transgenic plants, and the transcription level of the genes involved in nitrogen uptake or assimilation, including *NRT1.1*, *NRT2.1*, *NIA1* and *NIA2* was repressed. Therefore, the LBDs in *Alfalfa* are highly conserved with their orthologs in embryophytes. Our observations that ectopic expression of *MsLBD48* inhibited *Arabidopsis* growth by repressing nitrogen adaption suggest the negative role of the transcription factor in plant uptake of inorganic nitrogen. The findings imply the potential application of *MsLBD48* in *Alfalfa* yield improvement via gene editing.

## 1. Introduction

*Alfalfa*, a perennial forage legume, provides a high level of crude protein and plenty of digestible nutrients and mineral elements for livestock. It has become one of the most important forages globally due to its features of high yield, good palatability, nitrogen-fixation and strong adaptability [1]. The cultivated *Alfalfa* is autotetraploid (2n = 4x = 32), allogamous and highly heterozygous, characteristics that have hindered breeders in generating cultivars with economically important traits. The recent publications of genome assembly of several *Alfalfa* including *Medicago sativa* spp. *Caerulea*, a diploid progenitor of the autotetraploid *Alfalfa* [2,3,4], will facilitate gene identification and the improvement of molecular breeding strategies of the forage crop.

The *LATERAL ORGAN BOUNDARIES DOMAIN* (*LBD*) gene family, known as the *asymmetric leaves 2* (*AS2*), is one of the plant-specific transcription factors which play critical roles in lateral organ development, pathogen response and secondary metabolism [5,6,7,8,9,10]. The LBD transcription factors are typically defined by an N-terminal LOB domain essential for DNA binding and protein interaction [11,12]. LOB domain generally comprises a conserved zinc finger-like motif named C block (CX2CX6CX3C), a GAS block containing glycine (G), alanine (A) and serine (S) consecutively, and a leucine zipper-like motif (LX6LX3LX6L) [13]. The LBDs have been identified in many plants including staple crops, forages and fruits, and each species possesses dozens of LBD paralogs, for example, *rice* (35), *Zea mays* (44), *barley* (24), *Lotus japonicus* (38) and *apple* (58) [7,14,15,16,17]. In *Arabidopsis*, LBD proteins were divided into two major classes [18,19]. LBD transcription factors in Class I tend to play key roles in development of the lateral organs, including leaves [18], roots [20] and inflorescences [21,22]. For example, *MtEFP1* and *LjLBD6*, *AtLBD6* orthologs from *M. truncatula* and *L. japonicus*, have been implicated in determination of lateral organ identity [23]. Functional studies of the Class II LBD proteins revealed their roles in secondary metabolism, hormone-mediated plant defenses, and assimilation of nitrogen nutrition [19,24,25,26]. Similar to *Arabidopsis*, soybean *GmLBD16* and *GmLBD23* from Class II-a were positive regulators related to plants [26].

As the engine of biomass production and thus of substantial economic interest, nitrogen (N) uptake/assimilation is an important physiological process for plant growth and development. In *Arabidopsis*, the nitrate transporter (NRT) family and nitrate reductases (NR) have been demonstrated to play a crucial role in nitrate (NO_3_^−^) uptake and assimilation. For example, *Arabidopsis* chlorate-resistant mutant (*chl1*) identified as *AtNRT1.1* mutation showed defects in nitrate transport [27,28]. *AtNRT2.2* functioned redundantly for NO_3_^−^ transport in roots [29]. In *M.truncatula*, the nitrate transporter MtNPF6.8/MtNRT1.3 mediated low-affinity nitrate transport by affecting primary root growth [30]. Recently, *MtNPF6.5*, *MtNPF6.6* and *MtNPF6.7* were identified as three orthologs of *AtNRT1.1* mediating nitrate and chloride uptake, and *mtnpf6.7* caused a decrease in shoot biomass [31]. After the nitrogen uptake from soil, nitrate reductase (NR) reduces the NO_3_^−^ to nitrite in plastid, the first step of nitrogen assimilation. Two NR isoforms (AtNIA1 and AtNIA2) were found in *Arabidopsis* and deficiency of NIA1 and NIA2 resulted in significantly advanced bolting and biomass decrease due to the reduction of NR activity [32]. A similar phenomenon was also observed in *Brassica rapa L.* [33]. Previous studies have demonstrated that *AtLBD37*, *38* and *39*, Class II LBD members of *Arabidopsis*, functioned redundantly in nitrogen uptake and assimilation by depressing the expression of *AtNRT1.1*, *AtNRT2.1*, *AtNIA1* and *AtNIA2* [5]. The mechanism of the transcription inhibitory activity was elucidated in rice based on the observations that *OsLBD37*, *OsLBD38* and *OsLBD39* directly bound to the promoters of *OsNRT2.1*, *OsNRT2.2* and *OsNRT2.3* [34].

In this study, the LBD transcription factors in autotetraploid cultivated *Alfalfa* (*Medicago sativa* L.) were systematically identified and characterized. The framework of content in this study is shown in Figure 1. In addition, seven *LBD* genes were significantly involved in response to nitrogen signaling. Finally, we have described in detail the expression characteristics of *MsLBD48* and molecular genetic analysis suggested that *MsLBD48* is linked to nitrogen signaling. Taken together, our work will provide clues for further study of LBD protein functions and furnish the candidate gene for genetic engineering in *Alfalfa* molecular breeding.

## 2. Results

### 2.1. Sequence Identification of MsLBD Family Members and Distribution in Alfalfa Chromosome

To explore the potential members of LATERAL ORGAN BOUNDARIES (LOB)-DOMAIN-containing protein (LBD) in plants, including embryophytes and chlorophytes, the LOB domains of *Arabidopsis* LBDs were used as query sequences to BLAST against the database of 36 representative species (https://plants.ensembl.org, accessed on 12 September 2022). No hit was found in the five green algae we searched. In land plants, the number of LBDs varied from 22 in *Marchantia polymorpha*, a liverwort from a basal land plant lineage, to 90 in soybean (Figure 2), showing an increasing tendency of LBD members from non-vascular plants to higher plants. The results suggested that the LBD family evolved after plants had adapted to life on land.

For *Alfalfa*, the LBDs of *Medicago truncatula* acquired from the LIS Legume Information System (https://www.legumeinfo.org/, accessed on 12 September 2022) were used as query sequences against XinJiangDaYe, a Chinese *Alfalfa* landrace with genome sequence released in 2020 [3]. A total of 48 non-redundant LBDs encoded by 178 genes were identified in the autotetraploid genome. Among them, 174 MsLBDs were distributed on the 31 allelic chromosomes excluding Chr. 2.1, and the remaining four *MsLBD*s (i.e., *MS.gene041894*, *MS.gene011734*, *MS.gene040027* and *MS.gene40553*) were mapped to the scaffolds 32644, 32643, 24028 and 22534, respectively (Appendix A, Appendix A). Here, the 48 unidentical LBDs were named MsLBD1–MsLBD48 consecutively along the chromosome of *Alfalfa* (Appendix A). Meanwhile, analysis of LBDs of *M. sativa ssp. Caerulea*, the diploid progenitor of autotetraploid *Alfalfa*, revealed 46 members designated *M. sativa Caerulea LATERAL ORGAN BOUNDARIES (LOB) DOMAIN*-containing protein (MsCLBD1–MsCLBD46) distributing on the eight chromosomes particularly Chr. 5 and Chr. 8 (Appendix A, Appendix A). Although the putative LBDs varied substantially in length (142–318 amino acids in *M. sativa* and 86–318 amino acids in the diploid *Alfalfa*), the LBD transcription factors were predicted to localize in the nucleus (Appendix A).

Synteny analysis showed that most of the *LBD*s in the diploid and *Medicago truncatula* genome contain multiple orthologs distributed equally in four *Alfalfa* sub-genomes (Figure 3). Further analysis showed that the relative position of ortholog pairs in each sub-genome are the same and the sequences’ identity is extremely similar (95–100%). These results indicate that whole genome duplication (WGD) is a result of LBD family expansion in the *Alfalfa* genome. A total of 186 synteny gene pairs were identified between *M. sativa* and *M. sativa Caerulea*, and 176 gene pairs between *M. sativa* and *M. truncatula*, implying that *M. sativa Caerulea* that exhibit more gene pairs have a higher evolutionary relatedness than the *M. truncatula*.

### 2.2. Phylogenetic Relationship of MsLBD Family Members

Based on the phylogenetic analysis of LBDs from *Alfalfa* (48), *M. sativa* spp. *Caerulea* (46), barrel clover (57) and *Arabidopsis* (43), the LBD members are clustered into two classes (Class I and Class II) (Figure 4). Class I consists of five subclasses (I a–I e), and Class II contains two subclasses (II a and II b). Relative to Class II with a similar number of members for both subclasses, Class I accounting for about 85% of LBDs varied in size for the five subclasses with the majority (~60% of LBDs) belonging to I a, I b or I c (Appendix A). Within individual subclass, *Alfalfa* LBD members were highly conserved, while the homology identity dropped between subclasses, particularly the ones from Class I and Class II (Appendix A). LBDs from *M. sativa* spp. *Caerulea* showed a similar pattern of sequence homology (Appendix A). Also, the LBDs from the two perennial *M. sativa* (*Alfalfa* and *M. sativa* spp. *Caerulea*) were closer than their orthologs of *M. truncatula* (Figure 3), which is consistent with the kinship reported previously [2].

### 2.3. Sequence Features of MsLBDs and Gene Composition

As plant-specific transcription factors, LBD proteins contain a conserved LOB domain of approximately 100 amino acids at the N-terminus in a variety of plant species. Generally, the LOB domain has three blocks of amino acids including a conserved C-x(2)-C-x(6)-C-x(3)-C motif (C block), which is a defining feature for all LBD proteins, a glycine alanine serine (GAS) block and a L-x(6)-L-x(3)-L-x(6)-L (leucine zipper-like) domain [19]. Sequence alignment revealed that MsLBDs of Subclass I a, I b and I c shared the common features of LOB domain, while GAS block and leucine zipper-like domain varied for Subclass I d, I e and Class II members (Figure 5). Motif analysis of the 48 MsLBDs identified 10 motifs using MEME (https://meme-suite.org/meme/, accessed on 12 September 2022) (Figure 6 and Appendix A). Among them, motif 1 and 2, corresponding to GAS block and C block, respectively, were shared by all MsLBDs. Motif 4 residing between the GAS- and C-blocks was a common feature for Class I members, and motif 3 corresponding to leucine zipper-like domain was predominantly present in Class I excluding Subclass I e. For Class II MsLBDs, motif 8 and motif 5 are the counterparts of motif 4 and 3, respectively (Figure 6A,B). The replacement of the two motifs suggested functional divergency of the Class II MsLBDs from Class I members. In addition, motif 6 and 7 were found in Subclass I d, probably contributing roles specific to the subfamily.

Gene structure analysis showed that the number of exons for MsLBDs varied from one to three and the members from the same subclass tended to share identical exon–intron composition (Figure 6C). For example, most MsLBDs of subclass I a and I e consisted of a single exon. Subclass I b and Class II were composed of members with two exons, while exons of Subclass I c genes were separately by at least one intron. Exceptionally, Subclass I d covered all types of MsLBD with 1–3 exons regardless of the short length.

### 2.4. Cis-Acting Element Analysis of MsLBDs

In order to investigate the potential biological functions of the *MsLBD*s, the genomic sequence of 2000 bp upstream of the *MsLBD*s was extracted as a hypothetical promoter sequence for *cis*-acting element analysis. As shown in Appendix A, various types of *cis*-acting elements were found in the *MsLBD* gene family, including elements related to plant growth and development, hormones responses and abiotic stress responses. Promoter elements associated with growth and development were detected, such as GCN4 motif and AACA motif involved in endosperm expression, CAT motif related to meristem expression, and a large number of light-response elements such as Box 4, GA motif, G-box, and TCT motif. Additionally, a number of *cis*-elements associated with ABA, MeJA, auxin, and SA responses were identified, of which we found the motifs associated with the ABA response were the most abundant, followed by MeJA. Finally, we also found motifs related to stress responses, such as low temperature response, drought induction and anaerobic regulatory element (ARE) essential for anaerobic induction. Over all, various types of *cis*-elements were found in the promoter regions, suggesting that these *MsLBD*s may be involved in various biological processes and regulatory pathways.

### 2.5. Expression Patterns of MsLBDs

To compare the temporal and spatial expression profiles of *MsLBDs*, the transcriptome data (https://www.ebi.ac.uk/ena/browser/view/PRJNA276155?show%20=%20reads, accessed on 28 January 2023) were analyzed. Based on the standardized normalized FPKM values, 42 *MsLBDs* were detected to be expressed in at least one of the six tissues, i.e., leaf, flower, root, elongated stem, post-elongated stem and nodule, while FPKM of the remaining genes (*MsLBD15*, *16*, *28*, *35*, *39* and *43*) was not detectable due to the low abundancy of the transcripts in the selected organs (Appendix A). Notably, most *MsLBDs* of Subclasses I d and I e were undetectable, suggesting a very low abundance in the selected tissues. Although the expression level of *MsLBDs* varied in different tissues, it seems that some *MsLBDs* had a preference for certain tissue(s) and the paralogs of the same subclass shared similar expression patterns. For example, most members in Subclass I a had a relatively high expression in flowers. Subclass II members except *MsLBD3* were uniformly upregulated in nodules compared with other test tissues (Appendix A). The results that *MsLBDs* were expressed in a variety of temporal- and tissue-specific patterns suggests that these transcription factors may function in diverse processes.

### 2.6. Nitrogen Treatment Enhanced the Expression of Class II MsLBDs in Alfalfa

Given that Class II *MsLBDs* were expressed preferentially in nodules, we tested their transcription level in *Alfalfa* roots under the treatment of two inorganic nitrogen sources using qRT-PCR. Three batches of 4-week-old seedlings grown in a hydroponic system were exposed separately to KNO_3_ (3 mM), NH_4_Cl (3 mM) or KCl (3 mM) for one hour. Roots of the plants treated by the individual reagent were applied for gene expression analysis. The results showed that, compared with KCl (used as control), the transcript of *MsLBDs* was enhanced by both KNO_3_ and NH_4_Cl (Figure 7), and the increase by the former treatment was stronger than by the latter. In both cases, the transcript level of Class II a members (*MsLBD3*, *30* and *36*) increased at a lesser degree than that of the Class II b ones (*MsLBD1*, *5*, *29* and *48*). The findings suggested that Class II *MsLBDs* may play roles in assimilation of inorganic nitrogen sources, particularly nitrate in *Alfalfa*.

### 2.7. MsLBD48 Orthologs in Plant Kingdom

Previous studies have shown that Class II LBDs, such as *AtLBD 37*, *38* and *39*, and the corresponding orthologs in rice (*OsLBD37*, *38*, *39*) repress nitrate uptake [5,34]. Here, we focused on *MsLBD48*, the closest ortholog of *AtLBD37-39* (61.6%, 63.0%, 63.5%) in *Alfalfa* (Figure 4). In the plant kingdom, *MsLBD48* orthologs were found in land plants (embryophyta) including most vascular plants, and liverworts and mosses from bryophytes, but were absent in green algae (chlorophyte), suggesting that there were no *LBD48* homologs before plants evolved to live in terrestrial habitats. In angiosperm, monocots had several LBD48 paralogs, such as purple false brome (2), rice (3) and maize (4). Most eudicots except Rosid (e.g., *Eucalyptus grandis* and *Vitis vinifera*) and Brassicales (e.g., *Theobroma cacao* and *Gossypium raimondii*) possessed *LBD48* ortholog(s) with or without paralogs (i.e., legume species). The transcription analysis revealed that *MsLBD48* was expressed in both aerial and underground organs of *Alfalfa* at early flower stage and higher expression was observed in roots without nodules (Figure 8A). Also, the transcription level of *MsLBD48* was higher in mature shoots than the juvenile ones. The subcellular location analysis revealed that the MsLBD48-GFP fusion protein was localized to the nucleus when transiently expressed in the epidermal cells of tobacco (Figure 8B), which is consistent with the results of its orthologs in *Arabidopsis* and rice [5,6].

### 2.8. Over-Expression of MsLBD48 in Arabidopsis Repressed Nitrate Transportation and Assimilation

To investigate the biological functions, *MsLBD48* driven by CAMV 35S promoter was introduced into *Arabidopsis*. Compared with the wild type (Col-0) under the long-day conditions, the two independent transgenic lines overexpressing *MsLBD48* (OE1 and OE4) displayed pleiotropic growth defects, including dwarfism, pale-green leaves and early flowering (Figure 9A), which are reminiscent of the typical phenotypes of nitrogen (N) deficiency in plant [35]. Constitutive expression of *MsLBD48* was confirmed in the transgenic plants by semi-quantitative RT-PCR (Figure 9B). Measurement of chlorophyll content demonstrated that total chlorophyll in the MsLBD48-overexpression lines OE1 and OE4 was about 12% and 15%, respectively, less than the wild type (Figure 9C). In terms of days to bolting, both OE lines bolted at Day 18, about one week earlier than the wild type (Figure 9D). Calculation of leaf area (rosette leaves) revealed that the average size of mature rosette leaves from the transgenic lines was decreased to 27–30% of the wild type. Consequently, the fresh weight of the two overexpression lines was about one third of the non-transgenic control (Figure 9E). The retarded growth with pale yellowish-green plants of OE1 and OE2 indicated N deficiency in the *Arabidopsis* constitutively expressing *MsLBD48*. In *Arabidopsis*, members of the NRT family are the major transporters for absorption of nitrate from soil [36,37]. Otherwise, two isoforms of the nitrate reductase apoprotein, AtNIA1 and AtNIA2, have been reported to have a crucial role in nitrogen assimilation [32]. To investigate the molecular mechanisms underlying the control by *MsLBD48* of nitrate uptake and assimilation, we measured the expression levels of key genes (*AtNRT1.1*, *AtNRT2.1*, *AtNIA1* and *AtNIA2*) involved in nitrate absorption and assimilation. The results showed that, in the MsLBD48 expression plants, the transcription level of *NRT1.1*, *NRT2.1*, *NIA1* and *NIA2* was significantly reduced to about half of the wild type (Figure 9G–J), indicating that ectopic expression of *MsLBD48* resulted in the repression of the N metabolic genes.

## 3. Discussion

The LATERAL ORGAN BOUNDARIES (LOB) domain family proteins (LBD) are plant-specific transcriptional factors implicated in diverse processes of plant growth and development [24,37,38,39,40]. Based on the sequence analysis, there is no LOB ortholog in chlorophyte databases, implying that the LOB transcription factors are specific to the land-plants. Among the 33 representative embryophytes analyzed in this study, the number of LBDs ranged from 22 to 90 regardless of the genome size. For *Alfalfa*, the autotetraploid genome has evolved chromosome duplication from *M. sativa* spp. *Caerulea*, a diploid relative known as one of the ancestral species of *Alfalfa* [41]. In total, we have identified 48 LBDs encoded by 178 loci from 31 allelic chromosomes, and 47 LBD genes in *M. sativa* spp. *Caerulea*. The LBD genes seem to share high collinearity between *Alfalfa* and *M. sativa* spp. *Caerulea*. These findings are supportive of the genome duplication of the autotetraploid *Alfalfa* at chromosome scale derived from the diploid *M. sativa* spp. *Caerulea*. A conserved LOB domain is present in angiosperm including dicots and monocots [17,42,43,44,45,46,47,48,49]. The LOB domain consists of three motifs: cysteine motif, GAS-block and leucine-zipper-like motif. LBDs are generally composed of two main clades based on the leucine-zipper-like domain, which is present in Clade I but absent from Clade II members [43,48,49,50]. Our results showed all MsLBDs contained a conserved zinc finger-like motif, indicating structural and functional necessity. Two amino acid residues (E and G) after GAS block (Figure 4) are extremely conserved in MsLBDs suggesting that these two amino acids may have a special role in MsLBDs. Similarly, the evolutionary relationship of putative 48 redundancy MsLBDs were divided into two major classes consistent with previous studies on *L. japonicus*, soybean, rice and apple [7,14,17,26]. Gene structure and motif analyses further supported the phylogenetic tree indicating that the LBD gene family may be highly conserved among species.

The *cis*-regulatory elements and gene expression profiles can provide the important information about their biology function. Based on relative expression level, *MsLBDs* showed obvious tissue specific expression patterns (Appendix A) indicating diverse function among *MsLBDs*. A variety of response elements were found in the promoter regions, further supported this point. This inference is supported by LBDs’ functional diversity in regulating lateral organ development in plants [19,20,48,50]. The majority of the homoeologous genes shared similar expression patterns among different types of tissues (Appendix A), implying biological redundancy may exist across these *Medicago LBD* homologs. Previous studies have demonstrated that the *LBD* genes, especially Class II b members, were involved in regulating the nitrogen signaling response [5,34]. Similar to *Arabidopsis* and rice, the current study revealed that the *MsLBDs* in Class II showed differentially elevated expression patterns under additional nitrogen supplication (Figure 6). This result suggests that Class II LBDs may be a response to nitrogen availability in *Alfalfa* root. Two well defined transport systems (nitrate transporters and ammonium transporters) were found to deal with the nitrogen absorption from soil in plants [37,51]. Our findings revealed that higher relative expression levels of Class II b *LBDs* were induced by potassium nitrate compare to ammonium chloride suggesting a preference responsibility for nitrate in *Alfalfa*.

Nitrogen uptake and assimilation are essential growth-promoting processes in plants. Previous studies have demonstrated that the Class II members *LBD 37*, *38* and *39* have functional redundancy to repress nitrogen uptake at transcriptional level in *Arabidopsis* and rice [5,34]. *MsLBD48*, with a close evolutionary relationship and high degree of sequence identity to *AtLBD39* (Figure 3, Appendix A), may has similar functions to *AtLBD39* in nitrogen availability. We demonstrated that ectopic expression of *MsLBD48* suppressed the expression of nitrate uptake and assimilation associated genes *AtNRT2.1*, *AtNRT2.2*, *AtNIA1* and *AtNIA2* that caused many growth defects in *Arabidopsis* (Figure 8). These results suggest that *MsLBD48* may play a conserved role in nitrogen absorption as their homologs in suppressing NO_3_^−^ uptake. A protein’s subcellular localization pattern can provide significant clues to its function. Transcription factors target the nucleus to increase or depress the expression of downstream genes. Previous studies have demonstrated that LBD proteins in different plant species are localized to the nucleus [20,43,46,52]. We have demonstrated *MsLBD48* is located at the nucleus implying that *MsLBD48*, like other LBDs, functions as a transcription factor.

## 4. Materials and Methods

### 4.1. The Identification of MsLBD Gene Family in Alfalfa

The assembly genome files of *Medicago sativa* were downloaded from the figshare projects (https://figshare.com/projects/whole_genome_sequencing_and_assembly_of_Medicago_sativa/66380, accessed on 28 January 2023). To more comprehensively investigate the LBD family genes in the *Alfalfa* genome, the LBD protein sequences of *Arabidopsis* and *Medicago truncatula*, a model plant for legumes, were acquired from the database of *Arabidopsis* Information Resource (TAIR9) (www.arabidopsis.org) and the LIS Legume Information System (https://legumeinfo.org/, accessed on 28 January 2023), respectively. All these protein sequences were used as query sequences for BLAST analysis against the cultivar XinJiangDaYe genome database with an E-value cutoff of e^−5^. In addition, the *M. sativa* LBD proteins were further confirmed using a hidden Markov model (HMM) profile of LOB (PF03195) (http://pfam.xfam.org/, accessed on 28 January 2023) through HMME R 3.0 software with default parameters, and an E-value of 1.0 was set as the threshold. The lacked typical LOB domain protein sequences were filtered out with SMART (https://smart.embl.de/, accessed on 28 January 2023) and the redundant data were removed to produce a representative LBD gene set using CD-HIT software (https://github.com/weizhongli/cdhit, accessed on 28 January 2023) with default parameters. Subcellular localization of the MsLBDs were predicted with Cell-PLoc 2.0 (http://www.csbio.sjtu.edu.cn/bioinf/Cell-PLoc-2/, accessed on 28 January 2023), and the ExPASy ProtParam tool (https://web.expasy.org/protparam/, accessed on 28 January 2023) was used to predict protein physicochemical parameters [53].

### 4.2. Chromosomal Mapping and Syntenic Analysis

The MsLBD protein location was extracted from the annotation GFF file of ‘XinJiangDaYe’ and TBtools software was used to plot the gene chromosomal mapping situation. MCScanX was used to identify gene syntenic relationships among the LBD proteins of three different species, and the results were visualized using Dual Synteny Plot through TBtools (v2.0697).

### 4.3. Sequence Alignment and Phylogenetic Analysis

DNAMAN software was used for multiple sequence alignment of MsLBD protein sequences. The phylogenetic tree was constructed using the MEGA6.0 software neighboring joint method (Neighbor-Joining, NJ), and the bootstrap was set to 1000 times [54]. A conserved domain in the *Medicago sativa* LBD protein sequence was predicted using the online MEME tool (http://meme-suite.org/, accessed on 28 January 2023).

### 4.4. Gene Structure and Conserved Domain Analysis

Conserved amino acid sequences of LBD proteins were analyzed using the online MEME tool. MEME analysis parameters included a minimum width ≥ 6, a maximum width of 50, and a motif number of 10; all other parameters were set to default values. The intron–exon distributions of the *M. sativa* LBD genes were obtained using GFF annotation files from the ‘XinJiangDaYe’ *Alfalfa* genome. The results were visualized using TBtools software.

### 4.5. Cis-Regulatory Element Analysis

To analyze the cis-regulatory elements contained in *MsLBD* genes, the upstream 2000 bp genomic DNA sequences from start code of all predicted *MsLBD*s were separated from the cultivated ‘XinJiangDaYe’ *Alfalfa* genome and then submitted to the PlantCare database (https://bioinformatics.psb.ugent.be/webtools/plantcare/html/, accessed on 28 January 2023) to analyze the potential *cis*-regulatory elements in the promoter regions.

### 4.6. Transcription Profiling of MsLBDs in Different Tissues

The Medicago transcriptome data were downloaded from the CADL-Gene Expression Atlas (https://www.ebi.ac.uk/ena/browser/view/PRJNA276155?show%20=%20%20reads, accessed on 28 January 2023) provided by the Noble Research Institute. The FPKM data of six tissues, including roots, leaves, flowers, post-elongated stems, elongated stems and nodules, were generated by assembling the ‘XinJiangDaYe’ genome. The heatmaps of LBD expression levels were normalized and drawn by TBtools software.

### 4.7. Plant Material Growth Conditions and N Treatment

Seeds of *Medicago sativa* ‘zhongmu NO. 1′ were sterilized in 15% NaClO for 5 min, washed 3 times with distilled water and placed separately in glass Petri dishes with moist filter paper. Seeds were incubated in the dark for 48 h at 4 °C before being moved to room temperature to germinate. The 5 days germinated seedlings were grown in 1/2 Hoagland cultures for 4 weeks. For nitrogen treatment, seedlings were additionally supplied with 3 mM KNO_3_, NH_4_Cl and KCl in cultures for 1 h. The root samples were obtained and quickly immersed in liquid nitrogen for total RNA extraction. The 5 days germinated Zhongmu NO.1 seedlings were transferred to a vessel containing substrate with nutrient soil: vermiculite = 1:1 and kept growing in a greenhouse at 22 °C/18 °C, with a photoperiod of 16 h light and 8 h dark cycle for 7 weeks, to harvest the tissues (leaves, stem, root and flower). Tobacco seeds (*Nicotiana benthamiana*) were sowed on substrate (nutrient soil: vermiculite = 1:1) and grown in a greenhouse at 22 °C/18 °C, with a photoperiod of 16 h light and 8 h dark cycle for 5 weeks to analyze the subcellular localization.

### 4.8. Total RNA Extraction and qRT-PCR

For qRT-PCR, total RNA was extracted from samples using a plant total RNA extract kit (Promega, Beijing, China ). Preparation of cDNA was performed using the First-Script cDNA synthesis kit (Takara) according to manufacturer’s recommendations starting with 1 µg of total RNA. qRT-PCR was performed with the Light Cycler C1000 Real-Time PCR System (BioRad, Hercules, CA, USA); the primers are listed in Appendix A using *MsACTIN2* (*MS.gene013348*) as the housekeeping gene. The relative expression profile was calculated using the 2^−∆∆CT^ method.

### 4.9. Cloning of MsLBD48 and Construction of Transgenic Plants

The full length of the *MsLBD48* coding sequence was amplified from *Alfalfa* cDNA using primer pairs MsLBD48-f and MsLBD48-r, and PCR products were cloned into the T vector (TakaRa, Kusatsu, Japan) and sequenced. The full-length *MsLBD48* coding sequence was amplified from the T-MsLBD48 vector with primers MsLBD-OE-f and MsLBD48-OE-r, and subcloned into pCambia3301 plant expression vector to generate pCambia3301-*MsLBD48* for the overexpression study. The pCambia3301-MsLBD48 construct was introduced into *Arabidopsis* using a flower dipping method mediated by *Agrobacterium tumefaciens* (GV3101) [55]. Transgenic plants were selected on 1/2 MS plates supplied with 4 mg/L glufosinate ammonium.

### 4.10. Subcellular Localization Analysis

For subcellular localization, the *MsLBD48* full-length CDS removed stop codon was sub-cloned into pCamba1302 plasmid vector and then transformed to *Agrobacterium* strain (GV3101) using the freeze thaw method [56]. *Agrobacterium* strains carrying the recombinant plasmids were grown in liquid LB media supplemented with kana (50 mg/L) and rifampicin (50 mg/L). Cells were collected and resuspended in infiltration medium (OD600 = 1.0). The infiltration medium contained 5% sucrose, 50 mM MES, 2 mM MgCl_2_ and 0.1 mM acetosyringone. The resuspended culture was infiltrated to the tobacco leaves using injector with needle removed. The tobacco seedlings continued to grow for 48 h in the greenhouse and the leaf epidermises, peeled off from the leaf blades, were spread out on a microscope slide in water. The images were pictured with inverted fluorescence microscopy (Zeiss DP580, Jena, Germany).

### 4.11. Chlorophyll Content Measurement

Total chlorophyll content was determined as described by Liu [57]. Briefly, 0.5 g of 3 weeks seedling rosette leaves were extracted with 10 mL 100% ethanol for 24 h in the dark. The extracting solution were centrifuged at 12,000 rpm for 10 min at 4 °C, and absorbance was measured at 645 and 663 nm. Total chlorophyll concentration was calculated by Ca + b (mg/g FW) = 18.09 × A645  +  7.05 × A663.

## 5. Conclusions

This work identified and analyzed the structural and evolutionary relationship of the LBD gene family in *M. sativa.* Based on the sequence information of the 32 representative plant species, we found the LBD transcription factor family is present in the embryophytes but absent from the chlorophytes. We also characterized a group of Class II *MsLBD*s by nitrogen signaling responsibility using a qRT-PCR. Additionally, based on the analysis, we cloned and analyzed the expression pattern of the *MsLBD48*. Ectopic expression of *MsLBD48* in *Arabidopsis* showed significant effects on nitrate uptake and assimilation. This study will help to enable a deeper understanding of the structure–function relationships of LBD genes in *Alfalfa* and may enable new breeding techniques to improve *Medicago* yield production.

## Figures and Tables

**Figure 1 ijms-24-04644-f001:**
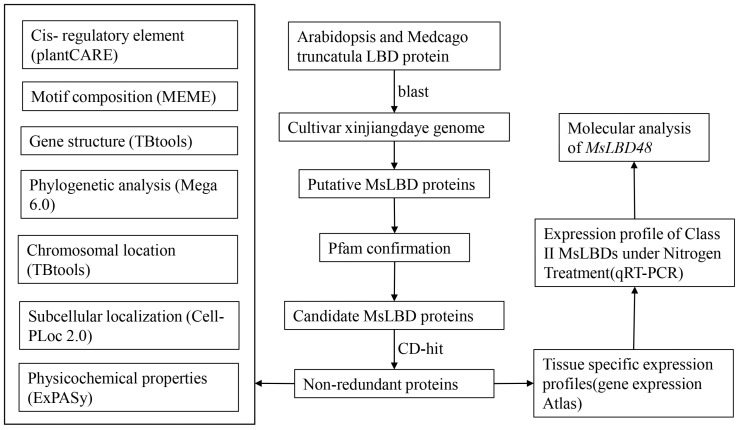
The framework of content in this study.

**Figure 2 ijms-24-04644-f002:**
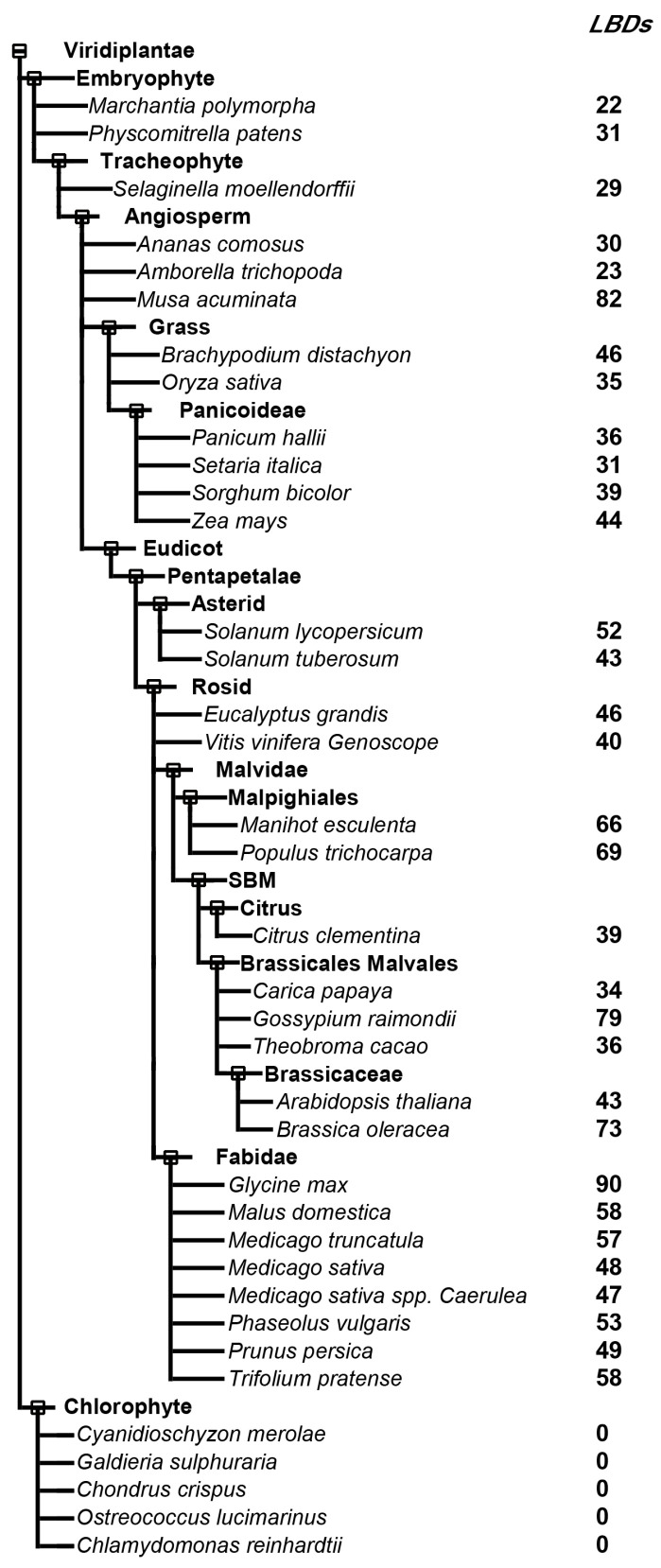
The number of LOB-domain like proteins in various plant species. The left panel shows a taxonomic tree of plant species in phytozomes from ensemble plants. The numbers indicate the LOB homologs.

**Figure 3 ijms-24-04644-f003:**
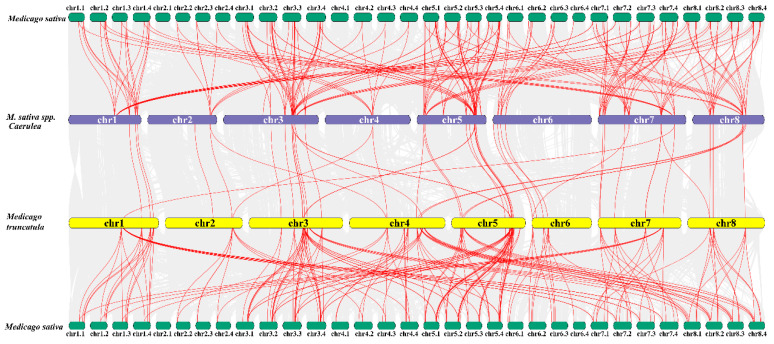
Gene synteny between *Alfalfa*, *M. sativa* spp. *caerulea* and *M. truncatula*. Red lines represent LBDs from the indicated species.

**Figure 4 ijms-24-04644-f004:**
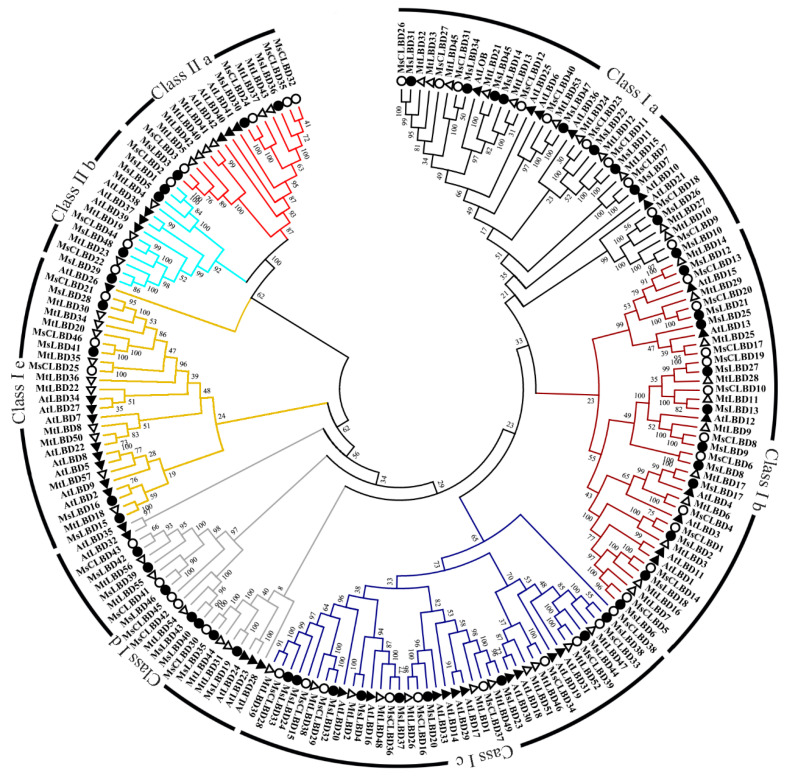
Phylogenetic analysis of LBD proteins in *Alfalfa*, *M. sativa* spp. *Caerulea*, *M. truncatula* and *Arabidopsis*. The tree was constructed by MEGA6.0 using the neighbor-joining method with 1000 bootstraps. The branch length represents the magnitude of genetic change. Individual LBD subfamily is represented by color.

**Figure 5 ijms-24-04644-f005:**
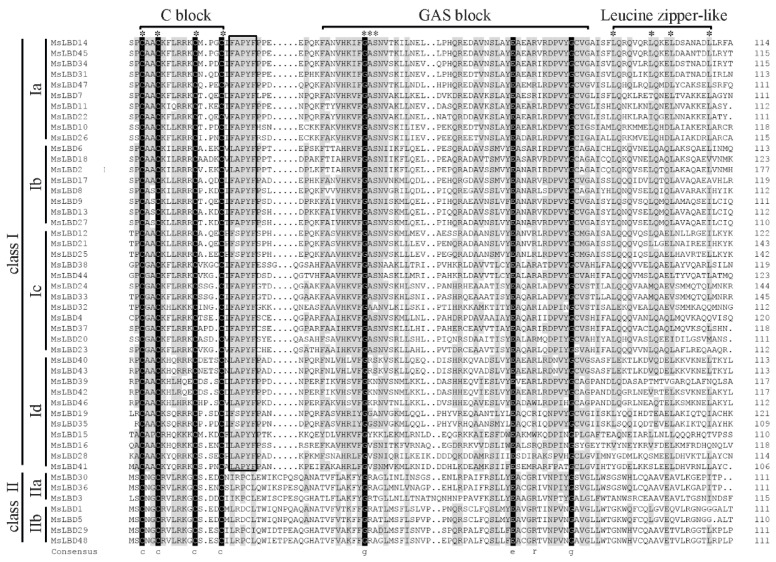
Alignment of the LOB domain of MsLBDs from *Alfalfa*. Asterisk (∗) indicated the key amino acids of LOB domain: cystine (C) of the C block; glycine (G), alanine (A) and serine (S) of the GAS block; and leucine (L) of the leucine zipper-like domain. Black box indicates (F/L)(A/S)PYF motif. Sequence identity >50% is indicated in grey, 100% in black.

**Figure 6 ijms-24-04644-f006:**
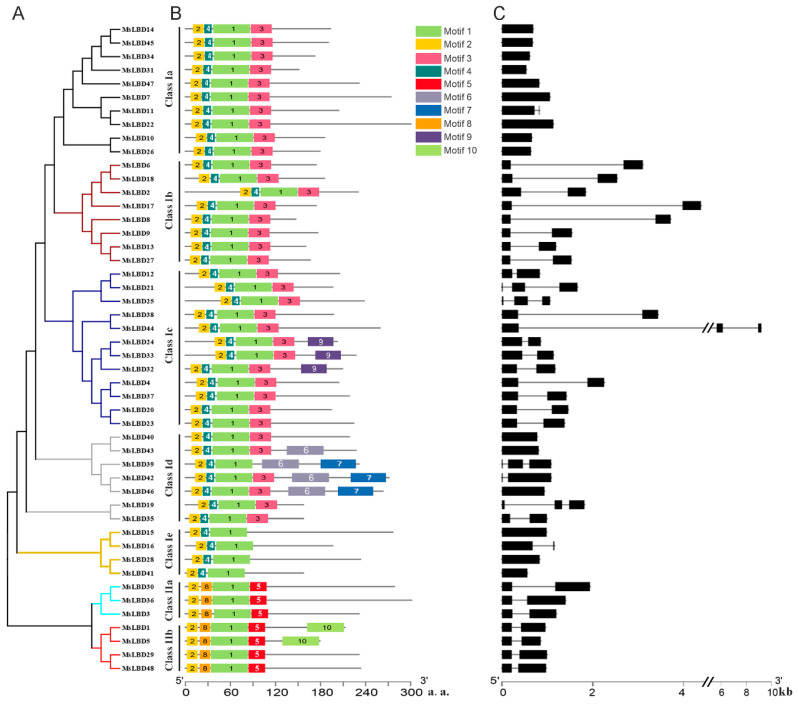
Features of 48 MsLBDs and their gene structure. (**A**) Homology analysis of MsLBDs from *Alfalfa*. The neighbor-joining (NJ) tree was constructed with full-length sequence using MEGA 7.0 with bootstrap value of 1000. Subgroup is indicated in different colors. (**B**) Distribution of the conserved motifs of MsLBDs. (**C**) Gene structure of *MsLBDs*. Exon and intron are represented by solid line and box, respectively.

**Figure 7 ijms-24-04644-f007:**
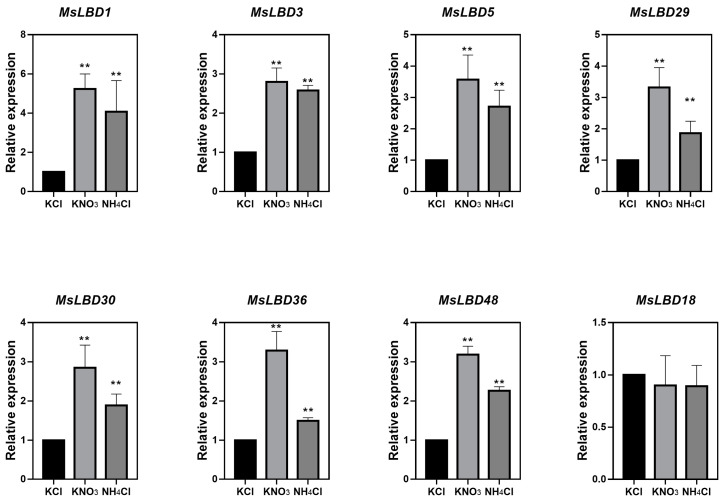
The expression analysis of Class II *MsLBD*s under KNO_3_ or NH_4_Cl treatment. Data were normalized to *MsActin 2* (*MS.gene013348*). Bars represented ±SD of three biological replicates. ** indicates *p* < 0.01 (Student’s *t*-test).

**Figure 8 ijms-24-04644-f008:**
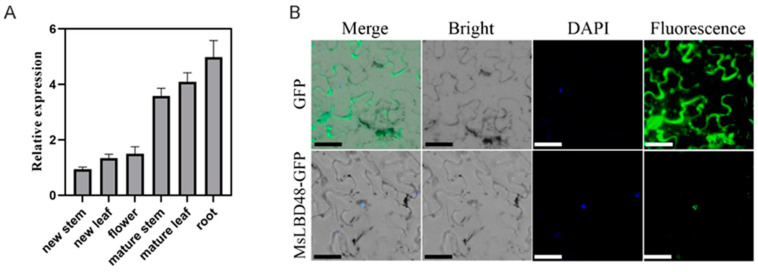
The expression profile of *MsLBD48*. (**A**). Expression analysis of *MsLBD48* in *Alfalfa* tissues using qRT-PCR. Bars represented ±SD of three biological replicates. (**B**). Subcellular localization analysis of *35S::MsLBD48*-GFP in epidermal cells of tobacco leaves transiently expressing the contract. Bar = 200 µm.

**Figure 9 ijms-24-04644-f009:**
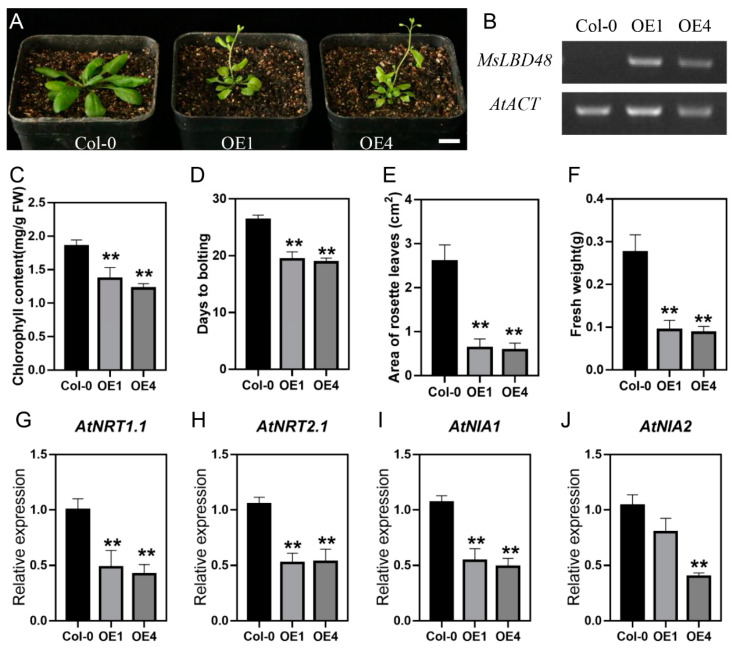
The *MsLBD48* gene is involved in the regulation of nitrogen uptake and assimilation-related gene expressions and affects N dependent status. (**A**) Image of the 19-day-old plants grown under LD. Bar = 2 cm. (**B**) RT-PCR detected the expression of *MsLBD48* in two *Arabidopsis* overexpression lines. (**C**–**F**) Phenotype of the *MsLBD48* overexpression lines. (**G**–**J**) The expression level analysis of key nitrate uptake and assimilation-related genes by qRT-PCR. Bars represent the ±SD of three biological replicates. The ** indicates *p* < 0.01 (Student’s *t*-test).

## Data Availability

Not applicable.

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
