# Peer review of "Genome-Wide Analysis of the LATERAL ORGAN BOUNDARIES Domain (LBD) Members in Alfalfa and the Involvement of MsLBD48 in Nitrogen Assimilation"

_ijms, 2023, doi:10.3390/ijms24054644_

Round 1

Reviewer 1 Report

This manuscript has identified 48 unique LBDs in Alfalfa and compared them with its diploid progenitor, M. sativa. In addition, ectopic expression of MsLBD48 in Arabidopsis indicated that MsLBD48 is a negative regulator in nitrogen uptake and assimilation. No discrepancies are found in the construction and logical development of the paper. However, this manuscript looks too preliminary and there is no significant novel finding, because authors merely examined the same role of LBDs known for Arabidopsis and rice in Alfalfa. In particular, overexpression analysis of MsLBD48 should be performed in Alfalfa, not Arabidopsis, because nodulation is unique in Alfalfa, not in other plants like Arabidopsis and rice. Ectopic expression sometimes induces any unexpected artificial results.

Reviewer 2 Report

In the present study, the authors described the structural and evolutionary relationship of the  LBD gene family in Alfalfa. In particular, investigated the MsLBD48 effects on nitrite uptake and assimilation. Overall the results will help to understand the relationships of LBD genes in alfalfa.

A framework figure is required. It will be useful to the readers for a better understanding of the studied issue.

Various cis-acting elements either induce or inhibit gene expression. So I suggest predicting the cis-acting elements in promoters of all the identified genes. It also validates by BIFC and yeast one-hybrid (Y1H) assays.

Reviewer 3 Report

The manuscript "Genome-wide Analysis of the LATERAL ORGAN BOUNDARIES Domain (LBD) Members in Alfalfa and the Involvement of MsLBD48 in Nitrogen Assimilation" by authors Xu Jiang, Huiting Cui, Zhen Wang, Junmei Kang, Qingchuan Yang, Changhong Guo is devoted to the consideration of a large family proteins in alfalfa. The manuscript is well written and contains interesting results. Especially, the revealed connection of one of the LBDs proteins with nitrogen metabolism. This is new data that will attract the attention of other researchers. In general, the work was done at a high level and can be recommended for publication in IJMS with minor revisions.

Minor shortcomings of this work is the lack of an obvious description of the connection and prospects of this study with the subject of the Special Issue. Also, the manuscript lacks a Conclusion section, which must be separated according to the rules of MDPI journals. In addition, in the last paragraph of the Introduction section, the authors present, as it were, already obtained results. This should be moved to the end of the Discussion section or merged with the Conclusion section. In the same paragraph, it is necessary to clearly indicate what goals the authors pursued when initiating this study.

Round 2

Reviewer 1 Report

I understand your claim and present situation.